# Task-Conditioned 3D U-Nets via Hypernetworks for Data-Scarce Medical Segmentation

**Luca Hagen**[1]  iD                                                    LUCA.HAGEN@FAU.DE
**Johanna P. Müller**[1]  iD                                  JOHANNA.PAULA.MUELLER@FAU.DE
**Moritz Gmeiner**[1]  iD                                          MORITZ.GMEINER@FAU.DE
**Bernhard Kainz**[1,2]  iD                                        BERNHARD.KAINZ@FAU.DE
[1] *Friedrich-Alexander University Erlangen-Nuremberg, GER*
[2] *Imperial College London, UK*

**Editors:** Under Review for MIDL 2026

## Abstract

Training 3D segmentation models typically requires extensive expert annotation, which is costly and often unavailable for rare or low-prevalence pathologies. We propose a hypernetwork-based framework that amortises the prediction of parameters for compact 3D U-Nets, enabling task-specific specialisation from as little as a single annotated volume. By learning shared anatomical structure, such as coarse shape, scale, and spatial organisation, across organs and imaging modalities, the hypernetwork generates task-conditioned network parameters, allowing controlled adaptation to previously unseen but anatomically related targets without full retraining. We evaluate the proposed approach on the CT TotalSegmentator and Medical Segmentation Decathlon benchmarks. The method achieves strong one-shot performance for anatomically homogeneous structures (*e.g.*, liver, spleen, atrium) and demonstrates stable few-shot adaptation for more heterogeneous or low-contrast targets (*e.g.*, tumours, prostate). In regimes with two to four annotated volumes, hypernetwork-generated U-Nets consistently outperform pretrained baselines and substantially reduce the performance gap to fully supervised models while using minimal annotation. These results indicate that weight prediction serves as an effective task-informed prior for data-scarce 3D medical image segmentation.

**Keywords:** Few-shot learning, Hypernetworks, 3D medical image segmentation, Data-Scarcity.

## 1. Introduction

Automated segmentation is a core component of many radiological workflows, with neural networks enabling accurate delineation of organs, tumours, and other anatomical structures. Architectures such as the U-Net (Ronneberger et al., 2015) and, more recently, Vision Transformer variants (Dosovitskiy et al., 2020) remain widely used for 3D medical image segmentation. Despite their success, these models typically rely on large amounts of expert-annotated data, which limits their applicability for rare pathologies or uncommon anatomical targets. The standard supervised learning pipeline requires extensive data collection, careful curation, and time-consuming expert annotation (Galbusera and Cina, 2024), followed by task-specific model training. While feasible for common and routinely imaged conditions, this paradigm breaks down when target cases are scarce. In such settings, the annotation burden represents a substantial entry barrier, despite the potential

clinical value of automation. One-shot and few-shot segmentation methods aim to address this challenge by conditioning a generic model on a small number of annotated reference examples. These approaches enable generalisation to unseen anatomical targets without retraining. Indeed, meta-learning based segmentation has already been explored in medical imaging (Khadka et al., 2022; Farshad et al., 2022; Leng et al., 2024; Alsaleh et al., 2024; Tirpude et al., 2025). More broadly, few-shot and low-shot learning in medical imaging has become an active research area, as reviewed in recent surveys (Pachetti and Colantonio, 2024; Dissanayake et al., 2025). However, despite their flexibility, these methods exhibit two fundamental limitations.

First, their performance typically remains below that of models trained specifically for a given task, even when additional annotated examples become available. Their ability to generalise across tasks often comes at the cost of limited capacity for task-specific optimisation. Second, fine-tuning is non-trivial, as task conditioning is often realised through shared internal representations or logit modulation rather than explicit parameter adaptation. As a result, updating model weights for a single task can degrade performance on others, limiting their usefulness when more data is acquired. Other lines of work attempt metric- or embedding-based segmentation under extreme scarcity, showing promising results even in one-shot scenarios (Cui et al., 2021). Approaches that rely on self-supervision and anomaly detection have also been proposed to mitigate the problem of limited foreground/background discrimination in few-shot settings (Hansen et al., 2022). These efforts demonstrate the growing recognition of data scarcity in medical segmentation and the need for more robust, adaptable methods. We address these limitations by formulating task conditioning as an explicit parameter-prediction problem. We propose a hypernetwork that amortises the generation of parameters for compact 3D U-Nets, conditioned on limited annotated data. Trained across a diverse set of segmentation tasks, the hypernetwork learns shared anatomical structure such as coarse shape, scale, and spatial organisation. By predicting weights rather than modulating logits or embeddings, the approach produces fully instantiated, task-specific U-Nets that are decoupled from the hypernetwork after generation. This separation allows the generated U-Nets to be fine-tuned using standard optimisation as additional data becomes available, without interfering with the learned task prior or other tasks. As a result, the proposed method combines the data efficiency of few-shot learning with the robustness and interpretability of conventional segmentation models, enabling controlled adaptation to anatomically related targets under limited supervision.

**Contributions.** This work makes the following contributions: (1) A hypernetwork framework that generates task-specific 3D U-Net parameters from minimal annotated data for one- and few-shot segmentation of unseen targets. (2) Task conditioning via explicit weight prediction, producing fully instantiated U-Nets that can be fine-tuned independently of the hypernetwork. (3) Demonstration that the hypernetwork captures shared anatomical structure across organs and modalities, providing task-informed priors for data-efficient segmentation. (4) Extensive evaluation on CT TotalSegmentator and Medical Segmentation Decathlon, showing strong one- and few-shot performance while reducing annotation requirements.

## 2. Background

Few-shot models have become a central strategy for medical image segmentation under data scarcity. Given only a few annotated reference volumes, these models adapt to new tasks by exploiting information extracted from the support set. Existing approaches can be broadly grouped into three paradigms. Similarity- and prototype-based methods classify query voxels by matching pixel-wise embeddings to prototypes constructed from the support set (Cui et al., 2021). While effective for anatomically homogeneous structures, their performance depends heavily on the expressiveness of the embedding space. As a result, they struggle with complex 3D anatomy, high intra-class variability, and the inherently difficult foreground–background separation in low-shot regimes (Cui et al., 2021; Hansen et al., 2022). Attention-based models use cross-attention to align query features to support features (Galbusera and Cina, 2024; Hu et al., 2025). This enables strong appearance transfer but tightly couples the inference procedure to the support examples. Since these models do not instantiate standalone segmentation models, they cannot be easily fine-tuned or deployed independently of the conditioning samples. Parameter-based meta-learning focuses on learning how to adapt a segmentation model from few reference examples. *MAML* (Finn et al., 2017) and its variants (Khadka et al., 2022; Leng et al., 2024; Alsaleh et al., 2024) learn shared initialisations that are rapidly fine-tuned to new tasks. Volumetric extensions show generalisation across heterogeneous targets (Farshad et al., 2022; Tirpude et al., 2025). However, because task adaptation is restricted to a few gradient steps, the resulting models remain bound to a local neighbourhood of the meta-initialisation, ultimately limiting their flexibility. First introduced in (Ha et al., 2016), hypernetworks generate the parameters of another model directly in a forward pass. They have been used for fast task-specific updates (Przewieźlikowski et al., 2024), full model parameterisations (Zhmoginov et al., 2022), and sample-conditioned filter generation (Nirkin et al., 2021). Unlike MAML-based methods, which rely on solutions near a shared initialisation, hypernetworks can represent a broader family of task-specific models. This distinction matters in few-shot 3D segmentation. MAML requires multiple fine-tuning steps, substantial per-task supervision, and operates on closely related tasks and organs. Our approach instead generates a complete task-specific U-Net in a single forward pass and applies only one light update on the generated model, leaving the hypernetwork unchanged. This enables stable, modular adaptation and effective generalisation to genuinely unseen anatomical targets from as little as one annotated volume. Motivated by these advantages, we adopt a hypernetwork to predict autonomous, fine-tunable segmentation models.

## 3. Method

To tackle data-scarce 3D medical image segmentation, we propose a hypernetwork framework that generates task-specific parameters for compact 3D U-Nets. Instead of feature- or logit-based conditioning, our method treats task adaptation as explicit weight prediction, producing fully instantiated U-Nets that can be fine-tuned independently. By training across diverse segmentation tasks, the hypernetwork captures shared anatomical structures, such as shape, scale, and spatial organisation, enabling adaptation to unseen but anatomically related targets with minimal annotated data. We first formalise the problem and then detail the architecture, training strategy, and task-specific parameter generation.

**Problem Formulation.** We focus on binary 3D medical image segmentation tasks. Each task $\mathcal{T}$ is defined by a dataset $D = \{(x_i, y_i)\}_{i=1}^{N}$, where $x_i \in \mathbb{R}^{H \times W \times D}$ is a 3D image volume and $y_i \in \{0, 1\}^{H \times W \times D}$ is its corresponding binary mask. For multi-class datasets with labels $y_i \in \{0, \dots, K\}^{H \times W \times D}$, we decompose the problem into $K$ binary segmentation tasks $\{\mathcal{T}_k\}_{k=1}^{K}$, one per foreground class. When volumes have multiple channels (e.g., different MRI sequences), each channel-class combination is treated as a separate binary task. Given a small support set of one or a few reference volume-mask pairs for a task $\mathcal{T}$, our goal is to predict the weights of a compact 3D U-Net that can be deployed as an autonomous segmentation model for that task. By formulating task adaptation as explicit parameter prediction, rather than feature- or logit-based conditioning, we aim to produce models that are both task-specific and fine-tunable without affecting other tasks.

**Target Architecture.** Our target network is a compact 3D U-Net composed of four encoder and four decoder stages, using a residual double-convolution block (ResDouble-Conv) as the primary building unit. Each block contains two 3×3×3 convolutions with group normalisation and LeakyReLU activations, alongside a 1×1×1 residual branch for stable gradient flow. Downsampling in the encoder is implemented via strided 3×3×3 convolutions, while upsampling in the decoder uses transposed convolutions. To reduce the number of parameters, skip connections between encoder and decoder stages are additive rather than concatenative. A final 1×1×1 convolution maps decoder features to segmentation logits. Across tasks, the encoder, normalization layers, and upsampling layers are shared and trained jointly. Task-specific adaptation is achieved by a hypernetwork that generates all convolutional weights in the decoder ResDoubleConv blocks and the final output head. This design produces compact, fully instantiated U-Nets with approximately 5.7 million trainable parameters, combining efficiency, interpretability, and the ability to fine-tune per task without affecting other tasks.

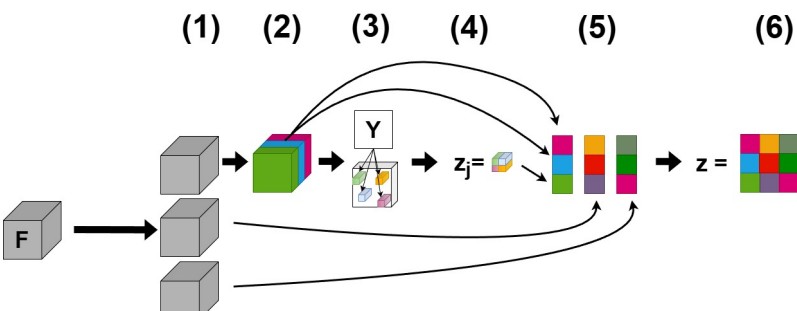

Figure 1: **Context construction**. (1) Create repetitions of input feature map $F$. (2) Group partitioning for each repetition. (3) Patch sampling from each group with respect to mask $y$. (4) A context vector $z_j$ corresponding to a base filter. (5) Stacking of all $z_j$ for one layer. (6) The context vector $z$ of the current layer.

**Hypernetwork-based U-Net Generation.** Given a reference volume-mask pair $(x, y)$ for a task $\mathcal{T}$, the hypernetwork generates all decoder weights of the corresponding target U-Net in a single forward pass. This allows the resulting U-Net to be fully instantiated and

fine-tuned independently of the hypernetwork. We first encode the input volume $x$ using the shared encoder $E$ to obtain a high-level feature map $F_{\text{enc}} = E(x) \in \mathbb{R}^{C \times H' \times W' \times D'}$. To summarise the task, we extract a global descriptor from the reference annotation by applying mask average pooling over the foreground region:

$$z_{\text{task}} = \text{AvgPool}\big(\text{Upsample}(F_{\text{enc}}) \odot y\big) \in \mathbb{R}^{1 \times C}, \tag{1}$$

where $\odot$ denotes elementwise multiplication, and $\text{Upsample}(\cdot)$ resamples the feature map back to the original resolution. This descriptor $z_{\text{task}}$ encodes the spatial and semantic information of the task and is subsequently used by the hypernetwork to generate decoder weights.

**Context Vector Construction for Decoder Convolutions.** To generate the weights of a decoder convolution, we build a compact context vector $z$ that summarises the current input feature map $F_{\text{in}}$, the global task descriptor $z_{\text{task}}$, and the support mask $y$. This vector provides the hypernetwork with both local and task-specific information required for weight generation. The entire process is visualised in Figure 1. The construction begins by aligning the support mask to the spatial resolution of $F_{\text{in}}$, so that each local patch can be labelled according to its value at the patch centre:

$$y_{\text{down}} = \text{Downsample}(y) \in \{0, 1\}^{h \times w \times d}. \tag{2}$$

Next, the channels of $F_{\text{in}}$ are partitioned into $G = c/C_{\text{base}}$ non-overlapping groups of size $C_{\text{base}}$:

$$F_{\text{in}} = \{F_{\text{in}}^{(i)}\}_{i=1}^{G}, \quad F_{\text{in}}^{(i)} \in \mathbb{R}^{C_{\text{base}} \times h \times w \times d}. \tag{3}$$

From each group $F_{\text{in}}^{(i)}$, we extract $N$ local $3 \times 3 \times 3$ patches

$$p_n^{(i)} \in \mathbb{R}^{C_{\text{base}} \times 3 \times 3 \times 3}, \qquad n = 1, \dots, N, \tag{4}$$

labelled according to $y_{\text{down}}$ at the patch centre. Sampling is balanced between foreground and background, with a fixed fraction $b$ of patches including boundary voxels to capture label transitions. Patches are ordered deterministically as positive $\rightarrow$ positive-boundary $\rightarrow$ negative $\rightarrow$ negative-boundary. Flattening and concatenating these patches produces the group-specific context vector:

$$z_i^{\text{context}} \in \mathbb{R}^{C_{\text{base}} \cdot 27 \cdot N}. \tag{5}$$

To generate all $G^2$ base filters of shape $(C_{\text{base}}, C_{\text{base}}, 3, 3, 3)$ for the convolution, the channel grouping and patch sampling procedure is repeated $G$ times with different partitions, yielding $G^2$ context embeddings that capture diverse local channel combinations. Each context vector is then augmented with a task-aware positional encoding. Each base filter is associated with a learnable positional vector $z_j^{\text{pos}}$, $j \in \{1, \dots, G^2\}$, which is combined with the global task descriptor using a two-layer MLP:

$$z_j^* = \text{MLP}\Big(\text{Concat}[z^{\text{task}}, z_j^{\text{pos}}]\Big). \tag{6}$$

Finally, the group-specific context vectors and their task-aware encodings are concatenated to form the final context vectors:

$$z_j^{\text{final}} = \text{Concat}\big[z_j^{\text{context}}, z_j^*\big], \tag{7}$$

which are then used by the hypernetwork to generate the decoder convolution weights.

**Hypernetwork.** The hypernetwork $H$ is implemented as a three-layer MLP with one hidden layer of dimension 2048 and GELU activations. It maps each final context vector $z_j^{\text{final}}$ to a base convolutional filter of shape $(C_{\text{base}}, C_{\text{base}}, 3, 3, 3)$. Layer normalisation is applied after the first two linear layers to stabilise training and improve convergence. To form full decoder layers, these base filters are stacked along the input and output channel dimensions to produce a convolutional kernel of shape $(C_{\text{out}}, C_{\text{in}}, 3, 3, 3)$, with $C_{\text{in}} = C_{\text{out}} = G \cdot C_{\text{base}}$. The weights and bias of the final $1{\times}1{\times}1$ segmentation head are generated by a separate two-layer MLP conditioned on the global task descriptor $z_{\text{task}}$. This design produces a fully instantiated decoder that is both task-specific and ready for independent fine-tuning.

**Reference-based weight update.** Once the decoder weights are generated, they are combined with the shared encoder to obtain a complete task-specific U-Net. To further leverage the information from the reference pair $(x, y)$, we perform a single gradient update on the U-Net parameters using a small learning rate, while keeping the hypernetwork fixed. The resulting model is a fully instantiated, task-specific U-Net that can be used directly for inference or as a task-informed initialisation for further fine-tuning when additional labelled data becomes available. This separation ensures that the hypernetwork retains its ability to generalise across tasks, while the generated U-Net can specialise to the current task.

---

**Algorithm 1** Episodic Training for Hypernetwork-predicted U-Nets

---
1: Initialize episode counter $episode = 1$
2: **while** $episode \leq N_{\text{episodes}}$ **do**
3:     Sample a mini-batch $\mathcal{B}$ of tasks
4:     **for** each task $\mathcal{T}$ in $\mathcal{B}$ **do**
5:         Sample a support pair $(x_s, y_s)$ for task $\mathcal{T}$
6:         Generate task-specific U-Net: $F_{\mathcal{T}} = \text{Hypernetwork}(x_s, y_s)$
7:         Sample an independent query pair $(x_q, y_q)$ for the same task
8:         Predict outputs: $\hat{y}_s = F_{\mathcal{T}}(x_s)$, $\hat{y}_q = F_{\mathcal{T}}(x_q)$
9:         Compute combined loss:
10:             $\mathcal{L} = \beta \, L_{\text{seg}}(\hat{y}_s, y_s) + (1 - \beta) \, L_{\text{seg}}(\hat{y}_q, y_q)$
11:     **end for**
12:     Update parameters of $E$, $H$, and shared decoder layers using AdamW with exponentially decaying learning rate
13:     $episode = episode + 1$
14: **end while**
15: Apply early stopping based on validation performance on unseen tasks

---

**Episodic Training Objective** We train the encoder $E$, hypernetwork $H$, and shared decoder components end-to-end in an episodic meta-learning setup. Each episode contains multiple tasks with support-query pairs. The training procedure is summarised in Algorithm 1.

## 4. Experiments

We assess our framework on tasks entirely unseen during training by generating a Hyper U-Net for each new target. Performance is compared against three baselines sharing the same U-Net architecture: (i) a conventionally trained, fully supervised model for the given target task (Conv U-Net; upper bound), (ii) a model using the shared encoder, upsampling, and normalisation layers but randomised decoder weights (Rand U-Net), and (iii) a model trained jointly on all tasks including target tasks (All U-Net) across both datasets, covering over 130 anatomical targets across CT and multiple MRI sequences. In addition, we report three ablation studies for our method, detailed in Appendix A.

**Datasets.** We train and evaluate on two public benchmarks selected for anatomical diversity and suitability for episodic meta-learning. CT TotalSegmentator (Wasserthal et al., 2023) provides voxel-wise annotations for over 100 thoraco-abdominal structures and is used to define a large set of binary training tasks spanning organs, vessels, bones, and soft tissue. The Medical Segmentation Decathlon (MSD (Antonelli et al., 2022) consists of 10 heterogeneous CT and MRI datasets covering diverse anatomies and pathologies. In MSD, all evaluation targets are defined as hold-out tasks that are never used during hypernetwork training and serve exclusively for out-of-distribution evaluation across both CT and MRI domains.

**Preprocessing and Training.** All volumes are resampled to isotropic $1.5\,\mathrm{mm}$ spacing and reoriented to the RAS coordinate system. CT intensities are clipped to $[-900, 900]$ and Z-score normalised, while MRI volumes are normalised by mapping the 1st to 99th intensity percentiles and clipping outliers. A single $128^3$ voxel patch covering the target structure is extracted per volume and centred when necessary. On-the-fly spatial (flips, $90°$ rotations, affine transforms) and intensity augmentations (Gaussian noise, smoothing, contrast adjustments) are applied. Models are trained for 20,000 episodes using a combined Dice and binary cross-entropy loss with $\beta = 0.2$, optimised with AdamW and an exponentially decaying learning rate. Early stopping is based on validation performance on unseen tasks. Training is performed on four NVIDIA A100 GPUs for approximately 12 hours.

**CT Domain.** First, we investigate the capabilities of our approach on unseen CT tasks. As our model was trained exclusively in the CT domain, this experiment examines the models' abilities to generalise to adapt to unseen organs / new tasks. We evaluate our hypernetwork on several tasks from the MSD dataset and report both the Dice Score (DSC) and the normalised surface distance (NSD) with a tolerance of $\tau = 2\,\mathrm{mm}$. As shown in Figure 1, a consistent trend emerges: Hyper U-Nets substantially outperform the randomised baseline across all tasks, yet remain well below the performance of a fully supervised U-Net. For several tasks the Hyper U-Net remain just below the fully supervised U-Net trained on all classes, even outperforming it on the hepatic vessels task (DSC 0.31 vs 0.05), The strongest results are observed for larger and anatomically homogeneous structures such as the liver and spleen (DSC 0.77 and 0.50; NSD 0.49 and 0.32), indicating that the hypernetwork captures coarse shape and spatial priors effectively. For more complex or fine-grained structures (e.g., pancreas, hepatic vessels), performance drops relative to fully supervised training but still clearly exceeds a random baseline. For highly heterogeneous targets such as lung and colon tumours, however, Hyper U-Nets provide only marginal gains over ran-

Table 1: **Organ- and Task-Shift results**. Dice (DSC) and Normalised Surface Dice (NSD) for Random U-Net, Conventional U-Net, and *1-shot* Hyper U-Net (Ours) across all CT tasks. *Models trained on target tasks. **Best** and second-best per task are highlighted.

| | Liver | | Spleen | | Hepatic Vessels | |
|---|---|---|---|---|---|---|
| **Network** | DSC | NSD | DSC | NSD | DSC | NSD |
| Rand. U-Net | 0.176±0.050 | 0.110±0.025 | 0.091±0.052 | 0.045±0.017 | 0.019±0.010 | 0.032±0.012 |
| All U-Net* | **0.826±0.073** | **0.668±0.106** | **0.580±0.167** | **0.509±0.123** | 0.050±0.023 | 0.038±0.114 |
| Hyper U-Net *(Ours)* | 0.772±0.063 | 0.490±0.080 | 0.501±0.169 | 0.316±0.115 | **0.307±0.120** | **0.364±0.129** |
| Conv. U-Net* | 0.951±0.018 | 0.792±0.069 | 0.911±0.032 | 0.708±0.108 | 0.607±0.104 | 0.731±0.103 |
| | Pancreas | | Lung Tumor | | Colon Tumor | |
| **Network** | DSC | NSD | DSC | NSD | DSC | NSD |
| Rand. U-Net | 0.016±0.007 | 0.017±0.006 | 0.006±0.014 | 0.005±0.006 | 0.012±0.017 | 0.010±0.010 |
| All U-Net* | **0.282±0.171** | **0.174±0.132** | **0.120±0.181** | **0.079±0.044** | **0.070±0.115** | 0.046±0.098 |
| Hyper U-Net *(Ours)* | 0.153±0.138 | 0.102±0.078 | 0.025±0.067 | 0.019±0.058 | 0.050±0.126 | **0.046±0.109** |
| Conv. U-Net* | 0.776±0.076 | 0.632±0.118 | 0.439±0.268 | 0.340±0.307 | 0.389±0.287 | 0.277±0.171 |

(a) Brain Edema

(b) Liver

(c) Hepatic Vessel

(d) Prostate

Figure 2: Qualitative evaluation on four target-tasks. Ground truth (red), and predictions of Hyper U-Net (blue) and Task-specific Conv. U-Net (green).

dom initialisation, indicating that in these settings a single support sample is insufficient to construct a meaningful task-specific model.

**MRI Domain.** In this experiment, in addition to a organ-/task-shift we introduce a modality shift. Our hypernetwork approach was trained exclusively on CT tasks, but is now evaluated on unseen organs in the MRI domain, presenting a severe challenge. For evaluation, we again use MSD tasks and report DSC and NSD. As shown in Fig. 2, the results mirror the CT experiments: Hyper U-Nets achieve strong performance on Heart and Brain Edema (DSC 0.50 and 0.33; NSD 0.28 and 0.22), demonstrating that the hypernetwork can generate meaningful parameters for targets unseen during training and even from an unseen imaging domain. Notably, they outperform the U-Net trained on all tasks on the Heart task (0.50 vs. 0.40). The overall pattern, however, remains consistent: for large, well-defined

Table 2: **Domain-, Organ-, and Task-Shift results.** Dice (DSC) and Normalised Surface Dice (NSD) for Random U-Net, Conventional U-Net, and **1-shot** Hyper U-Net (Ours) across all MRI tasks. *Models trained on target tasks. **Best** and second-best per task highlighted.

| Network | Brain Edema (FLAIR) | | Hippocampus (T1w) | |
|---|---|---|---|---|
| | DSC | NSD | DSC | NSD |
| Rand. U-Net | $0.002\pm0.002$ | $0.028\pm0.016$ | $0.002\pm0.001$ | $0.001\pm0.001$ |
| All U-Net* | **0.508±0.195** | **0.436±0.167** | 0.008±0.001 | 0.001±0.002 |
| Hyper U-Net *(Ours)* | 0.326±0.208 | 0.217±0.150 | **0.057±0.021** | **0.075±0.016** |
| Conv. U-Net* | $0.617\pm0.159$ | $0.584\pm0.109$ | $0.856\pm0.030$ | $0.970\pm0.034$ |
| **Network** | **Heart (bSSFP)** | | **Prostate (T2)** | |
| | DSC | NSD | DSC | NSD |
| Rand. U-Net | $0.022\pm0.005$ | $0.033\pm0.005$ | $0.004\pm0.005$ | $0.007\pm0.002$ |
| All U-Net* | 0.399±0.101 | **0.311±0.143** | **0.292±0.217** | **0.216 ±0.084** |
| Hyper U-Net *(Ours)* | **0.500±0.119** | 0.281±0.071 | 0.053±0.115 | 0.053±0.108 |
| Conv. U-Net* | $0.859\pm0.034$ | $0.619\pm0.111$ | $0.468\pm0.089$ | $0.364\pm0.056$ |

structures, Hyper U-Nets clearly exceed the random baseline, whereas for small or complex targets (e.g., hippocampus, prostate peripheral zone) they offer only marginal gains over random initialisation.

**Limited Data.** Further, we investigate how our Hyper U-Nets can be used when more than a single reference pair is available. Instead of treating the generated models as ready-to-use segmenters, we interpret them as task-informed initialisations. For a given number of reference pairs $N$, we construct a minimal train-test split in which 20% of the samples (at least one) are held out to assess generalisation. A Hyper U-Net is then generated from the training subset and subsequently fine-tuned on it using standard supervised training. Early stopping is employed to prevent overfitting. We report the resulting performance across several MSD tasks for $N \in \{2, 4, 8, 16\}$. The results (cf. Fig. 3) reveal an interesting trend: Hyper U-Nets can be effectively fine-tuned from as little as a single additional reference example, a setting in which conventional U-Nets typically fail due to overfitting. As expected, performance steadily improves as more data becomes available, and the Hyper U-Nets gradually approach the accuracy of fully supervised counterparts. Additionally, Table 3 compares our HyperU-Nets, generated from five reference samples, to a 5-shot MAML baseline (Alsaleh et al., 2024) on four abdominal tasks from the CT TotalSegmentator dataset. In Alsaleh et al. (2024), three of these tasks serve as meta-training tasks and the fourth as the held-out target, yielding split-dependent results. While MAML reports slightly higher scores on this small set of closely related organs, this setting is highly favourable for MAML due to the strong anatomical similarity between tasks, a regime in which MAML is known to perform well (Raghu et al., 2019). In contrast, our hypernetwork is trained across more than 100 heterogeneous tasks spanning multiple organs and modalities, and does not rely on such narrow task similarity.

| Model | Spleen | Liver | R Kidney | L Kidney |
|---|---|---|---|---|
| MAML (Alsaleh et al., 2024) | 0.839 | 0.903 | 0.775 | 0.870 |
| Hyper U−Net (*Ours*) | 0.821 | 0.865 | 0.753 | 0.801 |

Table 3: DSC of our method and an MAML approach using 5 reference pairs per organ.

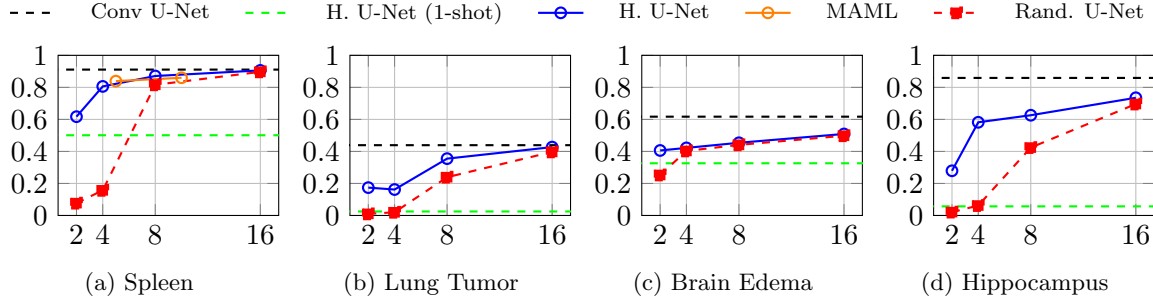

Figure 3: DSC of Hyper U-Nets (blue) and Random U-Nets (red) trained on $N \in \{2, 4, 8, 16\}$ samples across several tasks. A U-Net trained on the entire dataset (black) and a one-shot generated Hyper U-net (green) are reported as baselines.

## 5. Conclusion

In this work, we introduce HyperUNet, a hypernetwork framework that generates compact, task-specific 3D U-Nets from as little as a single annotated volume. By predicting decoder weights directly, rather than conditioning a shared backbone at the feature or logit level, HyperUNet produces fully instantiated segmentation models ready for immediate deployment or lightweight fine-tuning. This shifts task adaptation from gradient-based meta-learning towards weight-level conditioning and enables representation of a far broader family of task-specific models than approaches constrained to a single shared initialisation.

Experiments on CT and MRI tasks from the MSD demonstrate that one-shot weight generation is effective for anatomically homogeneous structures such as liver, spleen, and cardiac chambers, and that a hypernetwork trained only on CT generalises to MRI by capturing modality-robust anatomical priors. For heterogeneous or low-contrast structures, one-shot inference remains challenging; however, treating generated U-Nets as task-informed initialisations and fine-tuning on 2–4 labelled volumes consistently outperforms pretrained and randomly initialised baselines, closing much of the gap to fully supervised models.

Overall, HyperUNet provides a practical and powerful alternative to gradient-based meta-learning. Unlike MAML-style methods that require multiple fine-tuning steps on closely related tasks, HyperUNet generates a complete model in a single forward pass, with only a minimal weight update. This clean separation between shared priors and task-specific adaptation enables stable, modular specialisation to genuinely unseen anatomical targets under severe annotation scarcity. Future work will explore richer context representations, alternative encoders, and extension to additional modalities such as PET and ultrasound.

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

## Appendix A. Ablations

To understand the contribution of individual components in our Hyper U-Net workflow, we conduct a series of ablation experiments. Each subsection isolates one specific design choice and reports its effect on segmentation performance across multiple tasks of the MSD dataset, where our Hyper U-Net previously achieved strong results.

### A.1. Effect of Gradient Step on the Reference Example

In our workflow, after generating a Hyper U-Net from the reference volume–mask pair $(x, y)$, we additionally perform a single gradient update with learning rate $\eta$ on this same reference pair. This step further optimizes the generated model towards the reference example.
To assess the impact of this design choice, we compare the performance of Hyper U-Nets evaluated *before* and *after* applying the gradient step.

Table 4: **Ablation with and without gradient step**. DSC ↑ and NSD ↑ across all CT and MRI tasks. Best configuration per task in **bold**.

| Task | Model | DSC ↑ | NSD ↑ |
|---|---|---|---|
| Liver | w/o gradient step | 0.673±0.069 | 0.171±0.063 |
| | w/ gradient step *(Ours)* | **0.772±0.063** | **0.490±0.080** |
| Spleen | w/o gradient step | 0.476±0.165 | 0.289±0.130 |
| | w/ gradient step *(Ours)* | **0.501±0.169** | **0.316±0.115** |
| Hepatic Vessels | w/o gradient step | 0.281±0.102 | 0.306±0.100 |
| | w/ gradient step *(Ours)* | **0.307±0.120** | **0.364±0.129** |
| Pancreas | w/o gradient step | **0.186±0.137** | **0.123±0.076** |
| | w/ gradient step *(Ours)* | 0.153±0.138 | 0.102±0.078 |
| Brain Edema | w/o gradient step | 0.288±0.191 | 0.194±0.126 |
| | w/ gradient step *(Ours)* | **0.326±0.208** | **0.217±0.150** |
| Heart | w/o gradient step | 0.457±0.053 | 0.254±0.035 |
| | w/ gradient step *(Ours)* | **0.500±0.119** | **0.281±0.071** |

The DSC and NSD for both configurations are reported in Table 4. For DSC, we observe a consistent increase in performance across most tasks, with the largest gain for the liver (0.772 vs. 0.673). In contrast, the pancreas task shows a slight decrease when applying the gradient step.

For NSD, performance also improves for all tasks except the pancreas. Compared to the moderate DSC gains, NSD exhibits substantial improvements for the liver (0.490 vs. 0.171) and a notable increase for the hepatic vessels (0.364 vs. 0.306) when the gradient step is applied.

### A.2. Without normalization

The mask-averaged pooled encoder feature map $F_{\text{enc}}$ serves as our global task encoding $z^{task}$, which adapts positional encodings $z^{pos}$ and generates the weights and bias for the final $1 \times 1 \times 1$ convolutional layer.
We compare two Hyper U-Net variants: one employing normalization ($z^{task} = \text{LayerNorm}(\text{MAP}(F_{\text{enc}}))$) and one using the raw pooled features ($z^{task} = \text{MAP}(F_{\text{enc}})$).

Table 5: **Ablation with vs. without normalization**. DSC ↑ and NSD ↑ across all CT and MRI tasks. Best configuration per task in **bold**.

| Task | Model | DSC ↑ | NSD ↑ |
|------|-------|-------|-------|
| Liver | w/o norm | **0.844±0.056** | 0.411±0.070 |
| | w/ norm *(Ours)* | 0.772±0.063 | **0.490±0.080** |
| Spleen | w/o norm | 0.383±0.131 | 0.196±0.070 |
| | w/ norm *(Ours)* | **0.501±0.169** | **0.316±0.115** |
| Hepatic Vessels | w/o norm | 0.014±0.041 | 0.017±0.048 |
| | w/ norm *(Ours)* | **0.307±0.120** | **0.364±0.129** |
| Pancreas | w/o norm | 0.022±0.049 | 0.018±0.031 |
| | w/ norm *(Ours)* | **0.153±0.138** | **0.102±0.078** |
| Brain Edema | w/o norm | 0.015±0.036 | 0.025±0.239 |
| | w/ norm *(Ours)* | **0.326±0.208** | **0.217±0.150** |
| Heart | w/o norm | 0.053±0.130 | 0.137±0.025 |
| | w/ norm *(Ours)* | **0.500±0.119** | **0.281±0.071** |

We report the DSC and NSD of both configurations in Table 5. For liver and spleen, both relatively large and homogeneous organs, DSC and NSD remain of roughly similar magnitude across both configurations. For the remaining tasks, however, we observe a substantial degeneration of performance when no normalization is applied. On the hepatic vessels, pancreas, brain edema, and heart tasks, both DSC and NSD drop to values close to zero, indicating diffuse predictions that fail to capture underlying anatomical patterns.

### A.3. Task-dependent positional encodings

We hypothesize that enriching positional encodings with task-specific information enables the Hyper U-Net to assign task-dependent roles and importance to different convolutional filters in the final network.

Concretely, we define

$$z^* = \mathrm{MLP}\big(\mathrm{Concat}[z^{task}, z^{pos}]\big),$$

where $z^{task}$ is the global task encoding and $z^{pos}$ the positional encoding. To test this hypothesis, we compare two Hyper U-Net variants: (i) one employing task-dependent positional encodings $z^*$ as defined above, and (ii) one using purely positional encodings $z^* = z^{pos}$ without any task information.

The DSC and NSD for both configurations are reported in Table 6.
Across all tasks, using task-specific positional embeddings gives a higher DSC than using task-agnostic ones. The performance gain is most notable for Heart (0.500 vs 0.292) and Hepatic Vessels (0.307 vs. 0.130) and smallest for Brain Edema (0.326 vs 0.295). For the

Table 6: **Ablation with vs. without task-specific positional encodings**. DSC ↑ and NSD ↑ across all CT and MRI tasks. Best results per task in **bold**.

| Task | Model | DSC ↑ | NSD ↑ |
|------|-------|-------|-------|
| Liver | Regular PE | 0.654±0.151 | 0.158±0.055 |
| | Task-specific PE *(Ours)* | **0.772±0.063** | **0.490±0.080** |
| Spleen | Regular PE | 0.376±0.165 | 0.198±0.081 |
| | Task-specific PE *(Ours)* | **0.501±0.169** | **0.316±0.115** |
| Hepatic Vessels | Regular PE | 0.130±0.130 | 0.170±0.156 |
| | Task-specific PE *(Ours)* | **0.307±0.120** | **0.364±0.129** |
| Pancreas | Regular PE | 0.116±0.118 | 0.082±0.069 |
| | Task-specific PE *(Ours)* | **0.153±0.138** | **0.102±0.078** |
| Brain Edema | Regular PE | 0.295±0.197 | **0.246±0.132** |
| | Task-specific PE *(Ours)* | **0.326±0.208** | 0.217±0.150 |
| Heart | Regular PE | 0.292±0.130 | 0.204±0.059 |
| | Task-specific PE *(Ours)* | **0.500±0.119** | **0.281±0.071** |

NSD, we see a similar trend. The NSD is higher in almost all tasks, except the brain edema where the difference, however, is small (0.217 vs. 0.246). Similar to our previous ablation on the gradient step, we again see a signifant improvement of NSD for the Liver (0.490 vs. 0.158), Hepatic Vessels (0.364 vs 0.170) and Spleen (0.316 vs. 0.198) tasks.

