# OpenReview forum: "Task-Conditioned 3D U-Nets via Hypernetworks for Data-Scarce Medical Segmentation"
_MIDL.io/2026/Conference — MIDL 2026 Poster_

### Official Review · Reviewer_8dsT · 2026-01-08

**Confidence:** 3
**Preliminary Rating:** 4
**Final Rating:** 4

**Summary:**

The authors tackle the problem of few-shot/one-shot segmentation with hypernetworks, a meta-learning approach that learns weight modifications for pre-trained networks with limited manual annotation context. The authors demonstrate superior performance to general pre-trained baselines with less than 5 manual annotation examples.

**Strengths:**

The hypernetwork method is cleverly designed and makes sense on the top level. A single MLP able to translate context vectors into convolutional filters is a clever idea, and could raise interesting discussions for MIDL participants.

Validation is extensive, each target is represented by a generated binary network and the authors explored a large amount of targets.

This direction of hypernetworks could be more feasible than pre-trained gigantic general models, and this is supported by the reported results. Specially considering the generated networks can be further adjusted.

**Weaknesses:**

Figure 1 is weak, considering the complexity of the decoder weight generation, especially the context vector construction. I was not able to fully grasp how the context vectors are generated by looking at this figure and reading the description. The methodology appears to be very interesting, and in my opinion requires a large detailed figure illustrating every step, relating to the provided equations.

Comparisons are performed against a single reference method, although the authors mentioned others in the introduction/literature review.

**Detailed Comments:**

Its not clear to me how context vectors "know" which decoding layer they will map to (abstract, high number of channels, or close to the output, lower number of channels). Also, are the input features, including mask information, generated per UNet layer?

Any specific reason to CLIP HU intensities to -900 900 instead of, for example, -1024, 1024? Couldn't the 128³ patch not cover the structure border for large structures?

How was All U-Net trained with varying input and output channels?

Just a minor comment, It was not clear to me until the end that "few-shot" just means fine-tuning to N samples, not that your method is passive of extension to > 1 support pairs.

**Justification Of Final Rating:**

The authors have significantly improved the manuscript and addressed most of my questions. The new manuscript is easier to follow and understand, making it clear how the positional, context, and global features are calculated for the hypernetwork.

Other concerns remain regarding method comparisons. I understand lower results against supervised methods, but I missed more comparisons with similar meta learning methods. I still recommend acceptance as this is an interesting topic for discussion at the event.

**Justification Of The Preliminary Rating:**

This is amazing work. However, Figure 1 is in urgent need of improvement for better understanding of the methodology. Some of the points I raised in my questions need to be more clear, in my opinion, but that might be due to me not understanding.

**Questions To Address In The Rebuttal:**

The following questions are more a genuine curiosity than a critique of the work, and they might highlight topics that need to be more clear on the manuscript.

From my understanding, Algorithm 1 specifies how the hypernetwork is trained. It appears to me that each generation of a task-specific UNet is completely forward and at the "inference" level, however the encoder is always updated and kept "alive". Only the decoder is thrown away after inference. Can you confirm that?

Following that, we could say that the encoder is trained on all tasks, correct? Therefore, only half of the output network is generated and "meta-learning esque". Can you confirm and comment on the advantages and disadvantages? Could you apply this method to a strong encoder extracted from another method?

---

> ### Author Response · Authors · 2026-01-23
>
> We thank the Reviewer for the kind words and his reassuring feedback.
> Below, we clarify the open questions and configurations that may have been unclear.
>
> **Mapping context vectors to decoder layers.**
> The hypernetwork generates decoder weights sequentially in chunks, where each chunk corresponds to a specific part of a decoder layer. To ensure correct assignment, we attach a learnable positional embedding that encodes both the layer index and the channel position within the final architecture.
>
> **Layer-wise hypernetwork inputs.**
> Hypernetwork inputs are generated per layer. The shared encoder produces $F_{enc}$, which serves as input to the first decoder layer, and the output of decoder layer $i$ becomes the input to layer $i+1$. For each layer, the annotation mask is downsampled to the corresponding resolution, and boundary-/mask-aware patches are sampled from the current feature map. These patches form the hypernetwork input for that layer. The features $F_{in}$ arise naturally from the standard U-Net forward flow, with only patch sampling and mask downsampling being layer-specific.
>
> **CT windowing and patch sampling.**
> The CT window [−900,900] is a practical choice, although similar windows would also work. Empirically, it is important that sampled patches contain a mixture of foreground, background, and boundary regions; for our tasks and patch sampler, a patch size of $128^3$ was sufficient.
>
> **Partial-label training of the All U-Net.**
> The All U-Net is trained in a partial-label setting due to the use of multiple datasets with incomplete annotations. Instead of a binary segmentation head with sigmoid activation, it uses a 135-channel softmax head (134 classes plus background). For each training sample, we backpropagate only through the segmentation heads corresponding to classes annotated in the dataset from which the sample originates.
>
> **Shared encoder and generated decoder.**
> The encoder is shared across all tasks and kept fixed, while the decoder is generated on the fly at inference time and discarded afterwards (in practical applications, the generated decoder could be retained and fine-tuned for the specific task). Only the decoder follows a meta-learning paradigm. This design avoids the instability of generating the entire network from scratch and leverages the well-established benefit of strong shared encoders adapted via lightweight, task-specific decoders. As noted by the reviewer, this also enables seamless integration with pretrained encoders, which we view as an important direction for future work.
>
> **Clarity of context construction.**
> We agree that Figure 1 and the section describing context construction were unclear and updated them in the revision.

---

### Official Review · Reviewer_gD4w · 2026-01-10

**Confidence:** 4
**Preliminary Rating:** 2
**Final Rating:** 4

**Summary:**

This paper addresses the problem of few shot segmentation using a hypernetwork to generate the decoder weights of a UNet given a single volume-mask pair. This is a significant challenge in medical image analysis due to the resource-intensive nature of obtaining segmentation labels. Evaluation is performed on several tasks from TotalSegmentator and the Medical Segmentation Decathlon.

**Strengths:**

This appears to be a novel and inventive way of using hypernetworks for segmentations. One advantage of this method is that the result is a fully realized UNet architecture that can then be used or finetuned without the hypernetwork apparatus and there are some nice extra experiments demonstrating this.

**Weaknesses:**

There are two major weaknesses of this paper in my opinion.

The first is the suitability of the baselines. Unfortunately, the authors only compare against one other few-shot segmentation method (MAML), compared to which their method performs worse. However, there are many other such methods, as the authors discuss in their literature review. This is a major limitation of the paper. The "All - UNet" baseline is  appropriate to put the results in the context of a fully supervised upper bound, however I cannot discern the purpose of the "Rand - UNet" baseline, with random decoder weights. I would expect a network with randomised weights to perform very badly, and indeed it does. Beating this baseline is next to meaningless.

The second major weakness is that the particular architecture to create the hypernetwork input from the volume-mask pair is quite complex, involving breaking randomly sampling localised patches, but this complexity is neither well explained or justified. Within the time I can dedicate to MIDL reviews, I have tried to understand this method, but cannot fully due to various detail being unclear, for example:

- What is $F_{in}$? Which image is it calculated from, the test image or the example from the one-shot pair? Is it different to $F_{enc}$?
- The "position" embeddings appear to relate to channel indices, not spatial positions as it strongly implied by the name. It took me a while to figure this out...
- The hypernetwork appears to output the weights of a single filter at a time, rather than all filters simultaneously. However, this is not made clear until after equation 7, and it is necessary to understand this to make any sense of equations 2-6.

Unfortunately, I do not find the figure at all useful to understand the method. Equally importantly, why was method chosen over all other possible ways one could imagine using a hypernetwork here? To take one simple example, the input to the hypernetwork could simply be the $z_{task}$ plus a channel embedding (aka "position" embedding). Why is all this extra local patch complexity needed over this? it is not explained conceptually nor demonstrated experimentally.

**Detailed Comments:**

- Butoi's Universeg is not referenced here but is a highly cited paper in this
  space (far more highly cited than most of the other referenced papers). It
  seems like a strange omission from the references. See:

Butoi, Victor Ion, et al. "Universeg: Universal medical image segmentation."
Proceedings of the IEEE/CVF International Conference on Computer Vision. 2023.

- How is the example used in the "1-shot" experiments chosen? I would expect
  the performance to be quite sensitive to this input, and therefore it would probably be
  best to evaluate over a range of cases.

- I do not understand what the authors mean by "split-dependent results" when
  explaining the MAML results. This seems important because the MAML method
  outperforms this method.

**Justification Of Final Rating:**

The authors have significantly improved the paper since the initial submission in two important directions:
1. Though I think it could be further improved, the description of the method is much clearer than the initial submission
2. The authors have included in-context methods such as UniverSeg and Iris in the discussion, and the advantages of the proposed approach is much clearer to me and the reader.

In light of these improvements, I have improved my rating to "weak accept". I think this is an interesting line of work worthy of discussion at MIDL. However, I do still feel that the performance is a little underwhelming compared to other few-shot methods. I hope future work can build on this work.

**Justification Of The Preliminary Rating:**

Though I think this is an interesting and potentially fruitful line of work, it is difficult to recommend acceptance when the method is compared to only one other few shot segmentation method (and does worse). Furthermore, the clarity of the method would have to be improved significantly before acceptance.

**Questions To Address In The Rebuttal:**

- Conceptual explanation of the way the hypernetwork input is calculated is essential
- Clarification of the method along the line of some of the points I raise above.

---

> ### Author Response · Authors · 2026-01-23
>
> We sincerely thank the Reviewer for the extensive and constructive feedback. We address the main concerns below.
>
> **Purpose of the Rand U-Net baseline.**
> We agree that a U-Net with a randomly initialised decoder is expected to perform poorly and is not a meaningful benchmark to “beat”. We included Rand U-Net as a sanity check: since the encoder, upsampling, and normalisation layers are shared across all settings, its poor performance confirms that the gains of Hyper U-Nets come primarily from the generated decoder weights, not the shared components. We will clarify this explicitly in the revision.
>
> **Clarity and justification of the hypernetwork input construction.**
> We acknowledge that the current figure and description are difficult to parse. In the revision, we will improve clarity and explicitly note that $F_{in}$ is the input for the current layer to be generated; initially $F_{in}=F_{enc}$. Passing this through the generated layer produces $F_{out}$, which becomes $F_{in}$ for the next layer. The “position embeddings” encode channel/filter/layer position inside the decoder architecture, not spatial position, and we pointed this out in the revision to avoid confusion. Finally, the hypernetwork generates weights filter-wise (one filter at a time), which we also emphasised clearly in the revision.
>
> **Patch-based context vs. global task embedding.**
> Using only a global mask-pooled embedding ($z_{\text{task}}$) works on training tasks but generalises poorly to held-out tasks, indicating task-identifier shortcutting or “meta-overfitting”. To address this, we use a small set of meaningful local patches, including boundary-focused sampling, so the hypernetwork conditions on local anatomical evidence rather than memorising task identity. We added this rationale explicitly in the revision.
>
> **Channel embedding design.**
> While mixing channel groups may seem unintuitive, it is crucial for enabling interaction between channels initially assigned to separate groups. Without this, grouped convolutions would underperform given the task complexity.
>
> **Combining $z_{\text{task}}$ with $z_{\text{pos}}$.**
> Ablations show that this step substantially improves performance, suggesting that task-dependent spatial priors are beneficial in few-shot segmentation.
>
> **Reference selection sensitivity.**
> The one-shot reference example is sampled uniformly at random from the support split. We agree that sensitivity is important and reported mean $\pm$ standard deviation over eight randomly selected references in the revision.
>
> **Comparison to UniverSeg and other methods.**
> We thank the reviewer for pointing this out. UniverSeg and other ICL methods perform reference-conditioned inference via direct support-query interaction. In contrast, our method generates a standalone U-Net from a single reference that can be deployed without storing the support set and can be fine-tuned as new annotations become available. For this reason, MAML remains the most directly comparable baseline, but we included UniverSeg and similar approaches as baselines in the revision to strengthen the discussion.
>
> **Split-dependent results in MAML.**
> The MAML baseline was trained on three of the four tasks, with the remaining task held out as the test. To measure generalisation performance for each task, we trained the MAML baseline on the three remaining tasks. Therefore, each result corresponds to a separate training split, highlighting the split-dependent nature of these results.

---

### Official Review · Reviewer_rJR2 · 2026-01-10

**Confidence:** 4
**Preliminary Rating:** 3

**Summary:**

The paper proposes an alternative approach for few-shot medical image segmentation. It uses a hypernetwork to directly predict decoder weights of a (compact) U-Net conditioned on one or few annotated volumes. The key difference from prior work (MAML, prototype methods) is that task adaptation happens at the weight level rather than through gradient steps and feature modulation (light gradient update on the reference pair). The "generated" U-Net is fully decoupled from the hypernetwork and can be fine-tuned independently. The core idea is that this explicit weight generation can also act as a task-informed anatomical prior (for data-scarce medical imaging datasets) which is learned across many segmentation tasks. Evaluation on MSD with a CT-trained hypernetwork shows decent one-shot results for large homogeneous organs but poor performance on tumors and small structures. The method also transfers from CT to MRI with mixed success.

**Strengths:**

- The core idea of generating complete decoder weights in one forward pass is clear and technically well motivated.
- The decoupling property between the hypernetwork and the generated (standalone) U-Net is clean and has practical merit.
- The context vector construction (Section 3, Figure 1) has reasonable design choices (boundary-aware patch sampling, channel grouping and task-aware positional encodings). The ablations (Appendix A) systematically validate each component.
- The framing against MAML is reasonable. MAML needs task similarity and has second-order gradient costs. Hypernetwork appears to handle 100+ diverse tasks naturally without relying on strong task similarity. This is a genuine architectural advantage.
- The CT-to-MRI generalization experiment is ambitious. It is presented with both successes (heart) and failures (hippocampus) without overclaiming. Even partial success here suggests the learned priors are somewhat modality-agnostic.

**Weaknesses:**

- Limited novelty in core framework. Hypernetworks for few-shot weight generation exist in general vision - HyperSeg (CVPR 2021), Sylph (CVPR 2022), HyperShot (2023). The contribution is adapting this to 3D medical imaging with domain-specific design choices. Valid for MIDL, but raises the bar for experimental thoroughness.
- The hepatic vessels anomaly needs explanation. Hyper U-Net (one-shot, 0.307 DSC) beats All U-Net (trained on all 130+ tasks including hepatic vessels, 0.050). This is counterintuitive (full supervision should beat one-shot). That's a finding worth discussing. If it's a training issue with All U-Net, that baseline is unreliable.
- Some experimental details missing. The boundary patch fraction b is mentioned but never specified. Reference pair selection procedure not described. Is it random? If cherry-picked, results could be inflated.
- Tumor segmentation performs very poorly (lung tumor 0.025, colon tumor 0.050). This is acknowledged due to heterogeneity, which is fair, but what do predictions look like? All background? Diffuse predictions? Wrong locations? Understanding failure modes would help future improvements.
- The framing as anatomical priors (shape, scale, location) may explain why it inherently don't transfer to pathology (tumors have no consistent shape/location). This is a fundamental scope limitation. The approach may be inherently unsuited for pathology, which is where few-shot matters most clinically (rare pathologies).

**Detailed Comments:**

- Figure 3 needs clarification. X-axis shows N but with 20% held out, what does N=2 mean exactly? Training on 1 sample and testing on 1?
- The 3-layer MLP hypernetwork is simple given the task complexity. Was anything else tried?
- SAM-based few-shot methods are now standard. Even without full experiments, can you position your approach relative to them? A qualitative comparison or preliminary numbers on one task would help.
- The hepatic vessels result could be reframed positively as evidence against multi-task negative transfer (hurts multi-task models on thin structures).
- While MSD tasks are held out from training, targets like liver, spleen, pancreas are standard structures present in most segmentation datasets. Generalization claim would be stronger with anatomically novel structures not seen in any training distribution.

**Justification Of The Preliminary Rating:**

Solid application of hypernetworks to 3D medical few-shot segmentation. Design choices are sensible, ablations thorough and the authors have acknowledged the limitations. There is an architectural advantage over MAML (handling diverse tasks without second-order gradients). But core framework exists in general vision. The novelty is domain adaptation, which raises the bar for experimental rigor. More importantly, there's seems to be a mismatch between motivation (rare pathologies, data scarcity) and demonstrated success (common organs). The method learns anatomical priors that inherently don't transfer to pathology, limiting utility for the stated use case. Addressable with revisions, but the scope limitation may be fundamental.

**Questions To Address In The Rebuttal:**

- How stable are results across different reference pairs? Reporting mean±std over a small number of random selections (e.g. 5) would clarify if performance depends on lucky reference choice. High variance would be concerning.
- For tumor tasks, what do the predictions look like qualitatively? All-background, diffuse, or localized but wrong? All-background or diffuse suggests the hypernetwork can't generate meaningful weights for these targets. Localized but wrong suggests the anatomical prior actively misleads.
- The method succeeds on organs and fails on tumors. Do the authors view this as a fundamental scope limitation (anatomical priors don't apply to pathology), or a solvable problem? If the latter, what direction?
- What is inference time (generation + gradient step)? The efficiency claim over MAML would be stronger with numbers.

---

> ### Author Response · Authors · 2026-01-23
>
> We thank the Reviewer for the constructive feedback and for acknowledging the benefits of our approach. Below, we clarify the open questions and configurations that may have been unclear.
>
> **Missing parameters**
> The boundary patch fraction b is set to 0.7 in all experiments. We additionally evaluated other values and found that $b\in[0.5,0.8]$ yields comparable performance, with $b=0.7$ providing the best overall results.
>
> **Reference pair selection**
> Reference pairs are sampled uniformly at random. We split the dataset into disjoint support/query sets, sample one support volume to generate decoder parameters, and evaluated only on query volumes.
>
> **Clarity**
> In the revision we made several changes to improve clarity. In Figure 3 N denotes the number of available annotated samples (e.g. N=2 indicates two annotated samples). Consequently, for N = {2, 4, 8, 16} we get test-train splits of size [1, 1], [3,1], [6, 2] and [13, 3].
>
> **3-Layer MLP**
> While a 3-layer MLP is simple, it was chosen empirically: more complex hypernetworks performed well on training tasks but degraded strongly on held-out tasks, indicating “meta-overfitting”. We therefore use the 3-layer MLP as a compromise; future work may explore architectures that mitigate this effect.
>
> **Position relative to SAM and in-context learning (ICL)**
> SAM/ICL methods often predict masks relative to a reference via query-reference interaction. Performance can degrade with suboptimal references, anatomical variation, or domain shifts, and incorporating new labels is non-trivial. In contrast, our method produces absolute segmentations with a standalone U-Net and can be improved seamlessly via fine-tuning without relying on a fixed reference at inference. This relative positioning and line of thought was added to the revision.
>
> **Bad performance of All classes U-Net on hepatic vessels**
> Our All classes U-Net was trained jointly on 134 tasks with the same ~5M-parameter backbone, using a 135-channel softmax head (134 classes + background). Under partial labels (since we use different datasets), we backpropagate only for annotated classes and mask all others. We observe task-wise oscillations on difficult targets (e.g., hepatic vessels vs. lung tumour), likely due to gradient interference, limited capacity, and sparse supervision. The reported snapshot corresponds to the best overall checkpoint and is not always the one that maximises hepatic vessel performance.
>
> **Tumour predictions**
> In one-shot experiments, the main failure mode was incorrect localisation. In the lung tumour task, predictions often mirrored the tumour’s position in the reference, missing tumours elsewhere and sometimes segmenting similar-appearing structures instead. Performance also depended on tumour size: larger reference tumours led to better detection of similarly large tumours in the query. Higher image resolution consistently improved results for small structures, although we limited the voxel size to 0.8 mm³. For instance, Dice scores improved for colon (0.096 vs. 0.050) and prostate (0.168 vs. 0.053).
>
> **Fundamental scope**
> Our one-shot failures show that the hypernetwork generates weights focused on plausible distractors rather than diffuse predictions, indicating it overweights certain priors (e.g., location) that are often irrelevant for tumours. This is a solvable problem: explicitly weighting priors per task, topology, location, intensity, texture, self-similarity, etc., could guide the model to focus on relevant features, improving performance.
> Importantly, Figure 3 shows that even with limited data, our model outperforms a standard 3D U-Net on lung tumours, suggesting that reasonable priors are already encoded in one-shot predictions.
>
> **Baselines and Inference Time**
> In the revision we compared our approach to several related models on the MSD Pancreas Task. Inference time now is reported in the appendix.
>
> **Novelty**
> While hypernetworks have been used for weight generation, our work introduces a task-conditioned parameter prediction framework for data-scarce 3D medical segmentation, synthesising an autonomous, fine-tunable 3D U-Net from a single annotated reference volume. This differs from prior works:
> - **HyperSeg** (Nirkin et al.) predicts image-adaptive weights conditioned on the same input and targets intra-task/per-image variability.
> - **Sylph** (Yin et al.) generates lightweight classifier weights for incremental object detection, whereas we synthesise substantial parts of a dense 3D segmentation network (decoder + head) for voxel-wise prediction.
> - **HyperShot** (Sendera et al.) targets few-shot classification by generating a small classification module; in contrast, we generate a standalone 3D segmentation model that can be deployed and improved via standard fine-tuning as new annotations become available.

---

> > ### Comment · Reviewer_rJR2 · 2026-02-02
> > **Response to Rebuttal**
> >
> > Authors addressed surface concerns (details, baselines, clarity) and I appreciate the detailed response. But the fundamental issues remain - the method is motivated by rare pathologies, yet only works for common organs. Even if the tumour "improvement" is real (from 0.02 to 0.05 DSC) "outperforming 3D U-Nets", what clinical or research value does it provide? Marginal improvement between two failing numbers tells that this approach does not work for pathology. I find the authors' assertion that pathology priors are "solvable" is unsupported speculation.

---

> > > ### Author Response · Authors · 2026-02-02
> > >
> > > **On one-shot tumour performance and “clinical value.”**
> > >
> > > We agree that Dice scores in the 0.02-0.05 range are not clinically actionable, regardless of relative improvements. We do not claim clinical readiness for one-shot pathology segmentation; this limitation is stated in our discussion.
> > >
> > > **Motivation and scope under data scarcity.**
> > >
> > > While rare pathologies motivate the introduction, the paper’s goal is task-conditioned segmentation under extreme annotation scarcity with a standalone U-Net that can be fine-tuned as soon as more labels are available. As shown in our experiments, adding a few annotations yields strong gains: with 2 annotated volumes, performance rises to ~0.2 DSC, and with 8 volumes, we approach the supervised baseline (Lung Tumour). As addressed in the rebuttal, one-shot tumour failures are mainly mislocalisation/distractors, not diffuse or all-background collapse. This is consistent with tumours having weak location priors; a concrete future direction is to downweight location and emphasise appearance/texture priors per task, which, however, is out of scope for this work.
> > >
> > > Bottom line: one-shot tumours remain poor, but in the few-shot low-data regime, our hypernetwork shows a clear advantage over a standard 3D U-Net.

---

### Author Rebuttal · Authors · 2026-01-23

**Rebuttal:**

We thank the Reviewers for their constructive and encouraging feedback. Below, we address the remaining questions and clarifications integrated into our revision.

**Experimental Details**
The boundary patch fraction is fixed at $b = 0.7$; values in $[0.5, 0.8]$ perform similarly. CT windowing $[-900,900]$ was used; similar windows work comparably. Boundary- and mask-aware patch sampling is critical, with a patch size of $128^3$ sufficient.

**Evaluation Protocol and Reference Selection**
Reference volumes are sampled uniformly from the support split; query volumes are segmented by the generated U-Net, and performance is reported only on queries. Twenty per cent of the $N$ annotated samples (rounded up, minimum one) are held out for evaluation (e.g., $N=2$: 1/1 train/test; $N=4$: 3/1). Fig.3 was clarified.

**Hypernetwork Design and Context**
Decoder weights are generated sequentially per layer with learnable embeddings encoding layer and channel identity. Hypernetwork inputs are constructed per layer from the standard U-Net flow using downsampled masks and boundary-aware patches. Global task embeddings alone generalise poorly, while local patches enforce anatomical reliance. Channel-group mixing and combining task and positional context are necessary. The 3-layer MLP was chosen empirically to avoid overfitting. Fig.1 now shows architectures and training flow explicitly.

**Baselines**
Rand U-Net confirms gains stem from generated decoder weights. The All-classes U-Net uses a 135-way softmax and partial-label training; task-wise oscillations arise from gradient interference, not a single class. MAML results are split-dependent. SAM-style and in-context learning baselines were added in Tab.3.

**Tumour Segmentation and Scope**
One-shot tumour failures mainly reflect localisation errors and overreliance on coarse anatomical priors. Increasing resolution improves small-structure performance. Especially strict low-data settings, HyperUNet consistently outperforms standard 3D U-Nets, showing that meaningful priors guide segmentation beyond memorising reference locations.

**Positioning and Novelty**
Unlike reference-conditioned or in-context methods, our approach generates a standalone 3D U-Net producing absolute segmentations and allowing seamless fine-tuning as new annotations arrive. We clarified these distinctions, reported inference time, and improved figure clarity in the revision.

*NOTE: Without highlighting changes, the revision has 12 pages*

**Supporting Material:**

/attachment/b83f3b48eb5088204f9a2d3997d0bf3f154aed2b.pdf

---

> ### Comment · Reviewer_8dsT · 2026-01-26
>
> Could the authors also provide the revised version without the red deletion indications? Given that significant changes were made to the method's description, the clean version would help.

---

> > ### Author Response · Authors · 2026-01-27
> >
> > Dear Reviewer,
> > Thank you for the suggestion. Unfortunately, OpenReview does not allow us to upload an additional PDF at this stage. For improved readability, we therefore uploaded an alternative view of the same Revision to our GitHub (https://github.com/luca-hagen/HyperUNet) in which only newly added text is highlighted and removed text is omitted (as you requested).
> >
> > To be clear, this document does not introduce any new changes beyond the revision already submitted on OpenReview; it is only a different presentation to make the edits easier to inspect.

---

### Comment · Area_Chair_jDYP · 2026-02-01
**Please update your final rating**

If you haven't done so yet, please don't forget to update your final rating by clicking “Edit” → “Official Review” by February 1st 2026 (23:59 AoE). Thank you for contributing to the review process.

---

### Meta-Review · Area_Chair_jDYP · 2026-02-09

**Recommendation:** Accept (Poster)
**Confidence:** 3

**Metareview:**

All reviewers appreciated the authors' rebuttal and believe this work will be valuable for discussion. They agree that the manuscript is now clearer.

---

### Decision · Program_Chairs · 2026-02-13

Accept (Poster)